# Live2Diff: Live Stream Translation via Uni-directional Attention in Video Diffusion Models

## Abstract

Large Language Models have shown remarkable efficacy in generating streaming data such as text and audio, thanks to their temporally uni-directional attention mechanism, which models correlations between the current token and *previous* tokens. However, video streaming remains much less explored, despite a growing need for live video processing. State-of-the-art video diffusion models leverage *bi*-directional temporal attention to model the correlations between the current frame and all the *surrounding* (i.e. including *future*) frames, which hinders them from processing streaming videos. To address this problem, we present **Live2Diff**, the first attempt at designing a video diffusion model with uni-directional temporal attention, specifically targeting live streaming video translation. Compared to previous works, our approach ensures temporal consistency and smoothness by correlating the current frame with its predecessors and a few initial warmup frames, without any future frames. Additionally, we use a highly efficient denoising scheme featuring a $KV$-cache mechanism and pipelining, to facilitate streaming video translation at interactive framerates. Extensive experiments demonstrate the effectiveness of the proposed attention mechanism and pipeline, outperforming previous methods in terms of temporal smoothness and/or efficiency.

## 1 Introduction

Large Language Models (LLMs) have recently been very successful in natural language processing and various other domains (Jiang et al., 2023; 2024; Touvron et al., 2023a;b; Vaswani et al., 2017). At the core of LLMs is the autoregressive next-token prediction, seamlessly enabling real-time streaming data generation. Such a mechanism processes data continuously as it streams in, bypassing the need for batch storage and delayed processing. This token prediction mode has been widely used in many applications such as dialog systems (Anthropic, 2024; OpenAI, 2024), text-to-speech (Wang et al., 2023a; Zhang et al., 2023b), audio generation (Copet et al., 2023; Huang et al., 2023; Kreuk et al., 2022), etc. Despite the recent success of this streamed generation of sequential data like text and audio, it has not been fully explored for another very common sequential data type: videos. However, generating videos in a streaming manner is clearly worth investigating given the growing practical demand, particularly in live video processing, where the original input frames need to be translated into a target style on the fly.

Motivated by this, we study next-frame-prediction for producing streaming videos, with streaming video-to-video translation as the target application. Most existing video diffusion models exploit temporal self-attention modeling in a *bi*-directional manner, where the models are trained on sequences of input frames to capture the pairwise correlations between all frames in a sequence (Blattmann et al., 2023a;b; Geyer et al., 2023; Guo et al., 2023; Gupta et al., 2023; Yang et al., 2023). Despite some promising results, these models have limitations for streaming video. Early frames in a sequence rely on information from later frames, and vice versa for those at the end, which impedes efficient real-time processing of each frame as it streams in.

To address this issue, we redesign the attention mechanism of video diffusion models for streaming video translation, ensuring both high efficacy and temporal consistency. First, we make temporal self-attention *uni*-directional via an attention mask, akin to attention in LLMs (Jiang et al., 2023; Vaswani

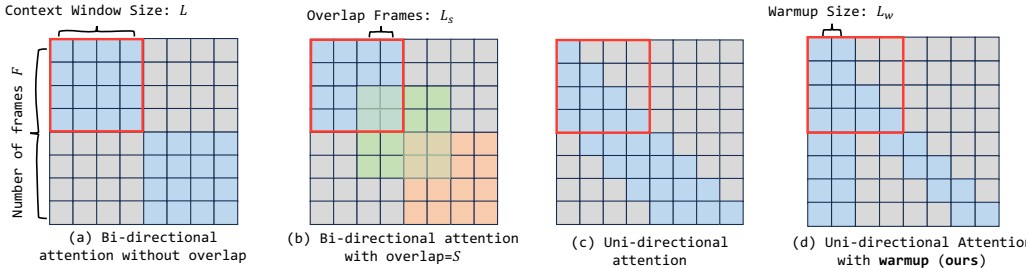

Figure 1: We visualize different types of temporal self-attention when the number of frames ($F = 8$) exceeds the length of the context window ($L = 4$). The $j$-th cell of the $i$-th row is highlighted if the output for frame $i$ may contain information from frame $j$. The red square delineates the attention mask used during training. (a) shows temporal self-attention in current video diffusion models, which is bi-directional within the context window without overlap between chunks. (b) uses a sliding window with overlap $L_s$ (three subsequent positions of which are highlighted in different colors, for clarity) and fuses the output of overlap regions. (c) denotes the uni-directional attention widely used in LLMs. (d) shows the attention proposed by our method. We set the initial $L_w$ frames as warmup frames and apply bi-directional attention to them, while using uni-directional attention for the subsequent frames. The initial warmup frames also contribute to the output for all future frames.

et al., 2017). This removes the dependency of early frames on later frames in both training and inference, making the model applicable to streaming videos. However, ensuring both high inference efficiency and performance on long video streams using unidirectional attention is non-trivial. An approach is to use dense attention with all previous frames for next-frame prediction. However, it increases time complexity and decreases performance when the video length exceeds the attention window size used during training. Another option is to limit temporal self-attention to a smaller fixed window size during inference. Unfortunately, unlike the sufficient context attention from user input tokens in LLM (Jiang et al., 2023; Vaswani et al., 2017; Xiao et al., 2023), it is difficult to generate satisfactory frames with limited context attention at the beginning of the video stream, which further results in artifacts in later frames. To tackle this, we introduce warmup area in the unidirectional attention mask, which incorporates bi-directional self-attention modeling to compensate for the limited context attention at the beginning of the stream. During inference, we include the attention from a few warmup frames at the start of the stream to the current frame. Such a tailored attention design ensures both stream processing efficacy and temporal consistency modeling.

Building upon our tailored attention mechanism, we present **Live2Diff**, a pipeline that processes **Live** video streams by a uni-directional video **Diffusion** model while ensuring high efficacy and temporal consistency. First, our attention modeling mechanism removes the influence of later frames on previous frames, allowing for the reuse of $K$ and $V$ maps from previously generated frames. This eliminates the need for recomputation when processing subsequent frames. We carefully designed a $KV$-cache feature in the diffusion pipeline to cache and reuse K/V maps, resulting in significant computation time savings. Second, we further include a lightweight depth prior in the input, ensuring structural consistency with the conditioning stream. Finally, Live2Diff uses the batch denoising strategy to further improve stream processing efficacy, achieving 16FPS for $512 \times 512$ videos on an RTX 4090 GPU. We conduct extensive experiments to validate the superiority of Live2Diff in terms of temporal smoothness and/or efficiency. We summarize our main contributions as follows,

- To the best of our knowledge, we are the first to incorporate uni-directional temporal attention modeling into video diffusion models for video stream translation.

- We introduce a new pipeline **Live2Diff**, which aims at achieving live stream video translation with both high efficacy (16FPS on an RTX 4090 GPU) and temporal consistency.

- We conduct extensive experiments including both quantitative and qualitative evaluation to verify the effectiveness of Live2Diff.

## 2    RELATED WORK

**Attention.**    LLMs (Jiang et al., 2023; 2024; Touvron et al., 2023a;b) owe their success largely to the remarkable efficacy of the attention mechanism (Vaswani et al., 2017). In order to support the auto-regressive prediction of the next token, they use a uni-directional (or "masked") attention mechanism, restricting the model to learning the dependence of *later* tokens on *earlier* ones, but no dependence of earlier tokens on later ones. However, for tasks with very long token sequences, relating the current token to all previous tokens becomes intractable. To address this, STREAMINGLLM (Xiao et al., 2023) proposes to relate the current token to several initial tokens and a number of most recent tokens, which improves efficiency in handling long tokens. While such kind of uni-directional attention is widely used in generating text and audio, video generation has not yet followed this trend: Bi-directional attention without masks is commonly used in video diffusion models (Blattmann et al., 2023a;b; Guo et al., 2023; Gupta et al., 2023) to generate video chunks. In this work, we study the use of uni-directional temporal attention in video diffusion models. While our method draws inspiration from STREAMINGLLM, it is the first time that such a design is studied in the video domain.

**Video Diffusion Models.**    The multitude of possible conditioning modalities has made diffusion models the basis for image editing approaches (Meng et al., 2021; Kawar et al., 2023), as well as video generation models (Guo et al., 2023; Liang et al., 2023; Kodaira et al., 2023). For example, ANIMATEDIFF (Guo et al., 2023) extends STABLEDIFFUSION by a so-called "motion module", enabling the denoising of entire video chunks based on temporal self-attention (Vaswani et al., 2017). FREENOISE (Qiu et al., 2023) is a method based on pretrained video diffusion models (e.g. ANIMATEDIFF (Guo et al., 2023)) for long video generation. This method carefully selects and schedules the latent noise for every time step in order to improve temporal smoothness. However, FREENOISE, according to their experiments section produces frames at under 3FPS on an NVIDIA A100 GPU, which is not acceptable in the kinds of live streaming scenarios that we aim at (see Section 1). FLOWVID (Liang et al., 2023) and RERENDER (Yang et al., 2023) produce frames at even lower rates, albeit with acceptably smooth results.

**Accelerating Diffusion Models.**    Some recent diffusion-based methods (Luo et al., 2023a;b; Song et al., 2023; Kodaira et al., 2023) have prioritized low latency and/or high throughput: (Latent) consistency models (LCMs) (Song et al., 2023; Luo et al., 2023a) have reduced the number of denoising steps from 50 (the default in STABLEDIFFUSION) to as low as 4, leading to large speed ups without too much loss in quality. This principle has even been combined with the use of low-rank matrices for fine-tuning (Luo et al., 2023b), allowing further speedup. A work that very specifically targets the streaming frame-by-frame translation setting is STREAMDIFFUSION (Kodaira et al., 2023): Not only is this technique utilizing the aforementioned low-rank-adapted LCMs, but also it denoises video frames in a "pipelined" manner for the streaming scenario, i.e. the batch of images to be denoised can contain different levels of remaining noise, allowing new frames to be added to the batch before previous frames in the batch have been completely denoised, which makes optimal use of GPU parallelization. However, STREAMDIFFUSION renders videos frame-by-frame without any temporal modeling, leading to significant temporal discontinuity, which our method avoids due to the temporal correlations learned during training.

## 3    METHOD

Our method, LIVE2DIFF, takes as input a stream of video frames, along with a matching text prompt. It produces a stream of output video frames, the spatial structure of which is similar to that of the input frames, but the appearance/style of which conforms to a specified target style, captured by DREAMBOOTH (Ruiz et al., 2023). To achieve this, we replace the bidirectional temporal attention used in previous approaches by *uni*-directional attention (Section 3.2). This allows us to cache $K$ and $V$ maps from previous frames, leading to increased throughput (Section 3.3). Furthermore we accelerate generation by pipelined denoising, i.e. multiple time steps with different levels of residual noise are denoised in parallel. By employing LCM-LoRA(Luo et al., 2023b) we can drastically reduce the number of necessary denoising steps, which also helps meet framerate criteria. We stabilize the spatial structure of frames with lightweight depth injection.

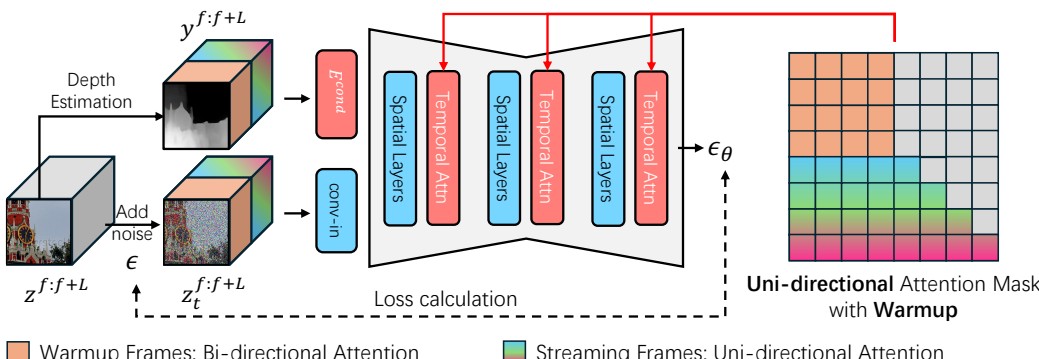

Warmup Frames: Bi-directional Attention          Streaming Frames: Uni-directional Attention

Figure 2: **The training pipeline of LIVE2DIFF.** During training, our model takes as inputs $L$ frames of noisy latents $z_t^{f:f+L}$ and depth conditioning $y^{f:f+L}$, where $f : f + L$ delimits the frame interval in a video stream, $t$ is the denoising timestep, $\oplus$ denotes point-wise addition. And we utilize a **uni-directional** attention mask with **warmup** to simulate the behaviour of streaming data.

### 3.1 PRELIMINARIES

**Diffusion Models.** Diffusion models (Ho et al., 2020; Dhariwal & Nichol, 2021) aim at undoing the so-called "forward process", that iteratively adds Gaussian noise to the representation of a sample from a distribution. To achieve this, STABLEDIFFUION (Rombach et al., 2022) trains a U-NET (Ronneberger et al., 2015) to estimate the noise component of a noisy latent representation of any given image. By repeatedly estimating remaining noise and removing (some of) this noise from the latent code, a purely Gaussian noise vector can iteratively be denoised to obtain a clean sample as follows: Given a noisy latent code $z_t$, the U-Net parametrized by weights $\theta$ computes the estimated noise $\epsilon_\theta(z_t, t, \mathcal{T}(c))$, where $\mathcal{T}(c)$ is the CLIP encoding (Radford et al., 2021) of a conditioning text string $c$. The less noisy latent code $z_{t-1}$ can then be approximated as

$$z_{t-1} \approx \lambda \cdot z_t + \mu \cdot \epsilon_\theta(z_t, t, \mathcal{T}(c)) \tag{1}$$

where $\lambda, \mu \in \mathbb{R}$ are constants derived from the noise schedule of the forward process (Song et al., 2020). The U-Net is trained by sampling images $x$ from the training distribution, mapping them to latent codes $z_0 = \mathcal{E}(x)$ and then adding varying amounts of Gaussian noise to obtain $z_t$, such that the U-Net output can be supervised by $L1$ distance to the known noise. Like in STABLEDIFFUSION, we use this as our main loss, but with $x$ holding not single images, but chunks of consecutive video frames.

**Bidirectional Attention in Video Generation.** Several video diffusion models (Guo et al., 2023; Blattmann et al., 2023b; Wang et al., 2023b; Chen et al., 2024) use bidirectional temporal self-attention layers to improve temporal smoothness of the output, essentially encouraging the model to learn temporal correlations. A temporal self-attention layer computes its output as

$$f_{\text{out}} := \text{softmax}\left(\frac{QK^\top}{\sqrt{C}}\right) \cdot V \tag{2}$$

where $Q := \mathcal{W}_Q \cdot f_{\text{in}}$, $K := \mathcal{W}_K \cdot f_{\text{in}}$, and $V := \mathcal{W}_V \cdot f_{\text{in}}$ are linear projections of the input features $f_{\text{in}}$ and $C$ is the number of feature channels. Absolute position encoding (i.e., sinusoidal position encoding) to $f_{\text{in}}$ before computing Eq. (2), to give the layer access to the temporal position of each feature vector. Once trained, the temporal self-attention module struggles to generate satisfactory results for frames that differ from the ones seen during training. Since previous works use such temporal attention layers without masking (Vaswani et al., 2017), $f_{\text{out}}$ can thus base its information about a particular frame on frames before and *after* that frame. Exploiting temporal correlations in this bidirectional way helps produce temporally smooth output, but is counter-productive for the streaming setting, as a prefix of $f_{\text{out}}$ will often need to be computed before the full $f_{\text{in}}$ is even available.

This bidirectional temporal attention design conflicts with two key requirements for streaming data inference: 1) the model must be able to handle frames of varying lengths, and 2) earlier frames

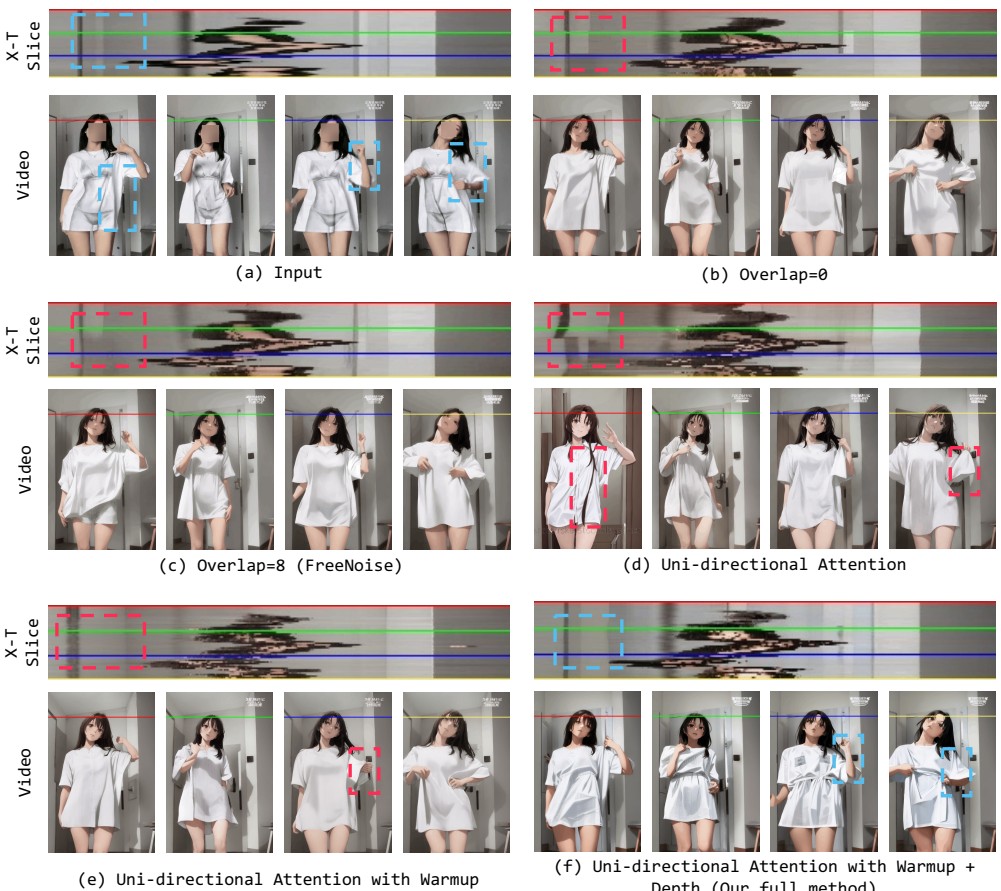

Figure 3: The X-T slice shows how the pixel values at the same X-coordinate change over time T. The position of the horizontal lines in the video corresponds to the X-coordinate positions visualized in the X-T slice. The color of each line represents the time in the X-T plot. Red dashed boxes denote regions suffering from flickering and structural inconsistency, while blue boxes indicate areas where these issues are resolved. Flickering and gradual change in the background region can be observed in (b), (c) and (d), which use the first three attention modes illustrated in Fig. 1 respectively. In case (e), with the last attention mode from Fig. 1 (see also Section 3.2, background flickering is reduced. The depth conditioning in (f) improves structure consistency further.

should not rely on information from later frames. To address these issues, some methods attempt to process video chunk by chunk (see Fig. 1 (a)) but this approach often results in abrupt transitions between chunks (Fig. 3 (b)). FREENOISE (Qiu et al., 2023) addresses the abrupt transition problem by introducing overlap between chunks and fusing the feature representations $f_{out}$ of the overlapping frames (see Fig. 1 (c)). However, this causes the overlapping frames to depend on information from later chunks, making it unsuitable for streaming input

## 3.2 UNI-DIRECTIONAL TEMPORAL SELF-ATTENTION WITH WARMUP

To turn bi-directional attention, as shown in Fig. 1 (a), in which each frame inside a chunk can be based on information from all other time steps in the chunk, into uni-directional attention, where each frame can only depend on *earlier* frames, we use masked attention (Vaswani et al., 2017; Jiang et al., 2023). Fig. 1 (c) presents a solution using the uni-directional attention mask commonly used in LLM(Touvron et al., 2023a;b; Vaswani et al., 2017; Jiang et al., 2024). However, as shown in red dashed box in Fig. 3 (d), the spatial structure of the first frame's output is inconsistent with the input, and the character identity differs from subsequent frames. Flickering in the background region can be observed in the X-T slice as well. We believe that the reason for this attention mode being less

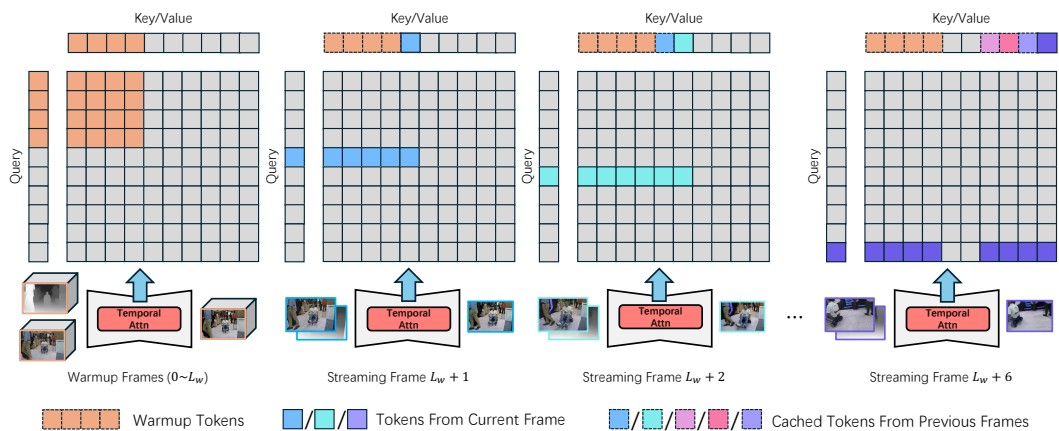

Figure 4: **Temporal attention during LIVE2DIFF inference stage.** During inference, we first input $L_w$ frames and apply bidirectional temporal attention, caching the $K/V$ in $KV$-cache. For the subsequent streaming frames, we compute the temporal attention using the cached tokens and add the $K/V$ to cache. If the number of cached tokens exceeds $L - 1$, we remove the earliest non-warmup cache from $KV$-cache.

effective for video transfer than for LLMs is rooted in the fact that in LLMs, attention operations rely on user prompts as initial tokens for uni-directional attention, while in video transfer such initial tokens first need to be generated by the model.

Based on this observation, we propose the attention mask shown in Fig. 1 (d). For the initial $L_w$ frames, we apply bidirectional attention to ensure video quality and stability during the warmup phase. For the following frames, we switch to a next-frame-prediction-based uni-directional attention, allowing the model to effectively handle streaming data. In training stage, as shown in Fig. 2, such attention mask is applied to all temporal attention modules. The results are shown in Fig. 3 (e). After trained with our attention mask, the flickering issue in the red dashed region has been resolved.

### 3.3 HIGH EFFICIENCY INFERENCE PIPELINE

In inference stage, similar to SDEDIT (Meng et al., 2021), we add a certain amount of Gaussian noise to the input frame(s), and denoise the noisy input to clean one with target style. We introduce $KV$-**cache** and **pipelined denoising** to establish a highly efficient inference pipeline in our method.

$KV$-**cache.** As described in Section 3.2 our warmup-based temporal self-attention makes sure that when we compute the attention for a certain frame, the attention for all previous frames has already been computed. This means that those parts of the matrices $K$ and $V$ in Eq. (2) that concern the previous frames do not need to be computed again, but can be retrieved from a cache.

Fig. 4 illustrates the behaviour of in our $KV$-cache for the simple case $L_w = 4$. In the first step, we feed $L_w$ frames into the U-Net and denoise them completely, with bidirectional attention as introduced in Section 3.2. This gives us the $K$ and $V$ matrices highlighted in orange in Fig. 4. They are cached and used for *all* future frames for temporal consistency. In the second step, the $L_w + 1$ (blue) frame arrives. We calculate temporal attention with cached warmup tokens and add the $K$ and $V$ of blue frame to $KV$-cache. The behavior in the third step is similar to that of the second step, with the only difference being that we utilize both the cached $K$ and $V$ pairs from the second frame and those from the warmup frames to compute the temporal attention. When video frames beyond the training context window arrive, we discard the cached frames that are not part of the warmup frames and are furthest from the current video frame. For instance, at $L_w + 6$ in Fig. 4, we discard the caches for $L_w + 1$ and $L_w + 2$, and perform temporal attention using the remaining cache with a length of context window $L$.

Note that the diffusion U-Net has multiple temporal attention layers, and that the U-Net needs to be applied $T$ times in order to fully denoise a frame. This means that for every combination of layer and denoising step, we keep a separate $KV$-cache.

**Pipelined denoising.** Similarly to STREAMDIFFUSION (Kodaira et al., 2023), we denoise frames in a pipelined manner, i.e. as soon as the next input frame becomes available, we add it to the batch of frames to be denoised, even though it may contain a much higher amount of noise than previous frames in the batch that have already undergone multiple denoising steps. This way we utilize our GPU capacity most efficiently, increasing throughput.

### 3.4 CONDITIONAL MODULE WITH STRUCTURE PRIOR

To facilitate the preservation of spatial structure we use an additional depth input: We use MIDAS (Ranftl et al., 2022; 2021) for frame-wise depth estimation. The depth frames are then encoded by STABLEDIFFUSION's encoder $\mathcal{E}$, with the results being fed into a lightweight convolutional module $E^{cond}$ (see Fig. 2). Finally, we add the output of the conditional module to the first convolution layer and pass it through the U-Net. We found this explicit structural prior to help the transferred video maintain better structural consistency with the source video. Evidence can be found in Fig. 3 and Section 4.

### 3.5 TRAINING

To train our model we use data collected from Shutterstock (Shutterstock, 2024), resized to resolution $256 \times 256$. We choose $L = 16$ and $L_w = 8$. We train our model as follows: We initialize the weights of our temporal self-attention modules with the weights from ANIMATEDIFF and fine-tune them for 3000 iterations using our uni-directional attention (Section 3.2). Then we add $E^{cond}$ (Section 3.4), with the last layer initialized with zeros (Zhang et al., 2023a) and train all weights jointly for 6000 iterations. We use the Adam (Kingma & Ba, 2014) optimizer with a learning rate of $1e-4$ and train on batches of 4 samples per GPU, on 8 GPUs. Accumulation of 32 gradients leads to an effective batch size of 1024.

## 4 RESULTS

### 4.1 EVALUATION SETUP

**Dataset.** We evaluate on the DAVIS-2017 (Pont-Tuset et al., 2017) dataset, which contains 90 object-centric videos. We resize all frames to $512 \times 768$ via bilinear interpolation and use COGVLM (Wang et al., 2023c) to caption the middle frame of each video clip. To specify the target style, we add the corresponding trigger words of DreamBooth and LoRA as suffix.

**Metrics.** We evaluate three aspects of the generated videos: *structure consistency* (Output frames should have similar spatial structure as input frames), *temporal smoothness* (no sudden jumps in the motion) and *inference latency*. We measure *structure consistency* as the mean squared difference between the depth maps estimated (Ranftl et al., 2021) for the input and output frames. As in previous work (Wu et al., 2023; Khachatryan et al., 2023; Guo et al., 2023) we measure *temporal smoothness* by CLIP score (Radford et al., 2021), i.e. by the cosine similarity of the CLIP embeddings of pairs of adjacent frames. In addition we compute the so-called "warp error" (Lai et al., 2018) for pairs of adjacent frames, i.e. we compute the optical flow (Teed & Deng, 2020) between the frames and then warp the predecessor frame accordingly, to compute a weighted MSE between the warping result and the successor. We also conduct a user study to evaluate *structure consistency* and *temporal smoothness*: Each participant is given triplets of videos (original input video, result from our method, result from a random different method) and is asked to identify the result with the best quality. Then we calculate the rate of our method winning compared to other methods. A higher win-rate indicates that users perceive our method to be better in the corresponding evaluation direction. Please refers to supplementary for detailed information. We measure *inference speed* as the total amount of time it takes each method to process an input stream of 100 frames at resolution $512 \times 512$, on a consumer GPU (NVIDIA RTX 4090).

### 4.2 COMPARISONS

We compare our method to three previous works, all based on STABLEDIFFUSION(Rombach et al., 2022) and compatible with DREAMBOOTH (Ruiz et al., 2023) and LORA (Hu et al., 2021): **STREAMDIFFUSION** (Kodaira et al., 2023) applies SDEDIT on a frame-by-frame basis. The same noise vector is used for all the frames, to improve consistency and smoothness. To achieve interactive

framerates, STREAMDIFFUSION uses LCM-LoRA(Luo et al., 2023b) and TINY-VAE(Bohan, 2023), with the latent consistency model scheduling (Luo et al., 2023a). We apply the same acceleration techniques in our method. RERENDER(Yang et al., 2023) first inverts input key frames into noisy latent codes. During denoising, temporal coherence and spatial structure stabilization are achieved by using cross-frame attention, flow-based warping and CONTROLNET (Zhang et al., 2023a). We select all frames as key frames for the purposes of our evaluation, but otherwise use the default settings. FREENOISE (Qiu et al., 2023) does not natively support an input video as conditioning, but by adding a sufficient amount of noise to the input, similar to our method and SDEDIT, we can nevertheless use it for our video-to-video translation task. The amount of noise we add is equivalent to half of the entire denoising process. FREENOISE uses a technique called window-based attention fusion, that (similar to bidirectional temporal attention) leads to frames incorporating information from future frames. This actually makes it unsuitable for the streaming setting, which we mitigate by giving FREENOISE access to *all* frames, not expecting to receive the first output frames after we have given the last input frames. In this sense we are giving FREENOISE a considerable advantage.

**Qualitative Comparison.** Fig. 5 compares outputs of all methods: While part (a) shows two consecutive output frames, part (b) shows frames that are further apart. STREAMDIFFUSION(Kodaira et al., 2023) exhibits strong flickering in the background (red box in (a)). When foreground and background are difficult to distinguish (box and shelf in (a), dog and table in (b)), works other than ours struggle to produce satisfactory results: STREAMDIFFUSION generates inconsistent results with low quality. RERENDER generates strong artifacts in the first frame (see (a)) and propagates them to later frames. FREENOISE fails to adhere to the input frame and generates elements unrelated to prompt and input. In contrast, our method leverages depth information to ensure the structural accuracy of the generated results (e.g. the box in Fig. 5 (a)) and maintains consistency over longer duration (dog in Fig. 5 (b)).

**Quantitative Comparison.** In Table 1 we evaluate structure consistency and temporal smoothness: While our method outperforms the others in structure consistency, we observe that FREENOISE achieves a better CLIP score and warp error for temporal smoothness. This is not surprising, as the way that time steps are correlated in FREENOISE allows information to flow bidirectionally along the temporal axis, which, unlike for all other methods, required FREENOISE to be given access to *all* frames at once (see first paragraph of Section 4.2). This is an unfair advantage to FREENOISE, violating some assumptions of the streaming scenario, as it allows FREENOISE to correlate its output frames to input frames that would likely not be available at the time the output frame needs to be produced. In our user study, all our win rates are way above 50% for both structural consistency and temporal smoothness, confirming that our results are the most convincing. STREAMDIFFUSION(Kodaira et al., 2023) scores second-best in structure consistency, likely because it applies only a moderate amount of noise to its input, but this limits its ability to conform with the target style (see also Fig. 5). Table 1 also compares inference latencies, i.e. the average time that goes by between receiving an input frame and producing the corresponding output frame. As is to be expected for a method that first consumes all the frames before producing any output, FREENOISE has by far the largest latency, which makes it unusable for the live streaming scenario. Only STREAMDIFFUSION has a better latency than our method, which we attribute to it making a different tradeoff between performance and quality. This is confirmed by the MSE, CLIP scores, warp error, and the user study results, that consistently indicate higher output quality for our method.

### 4.3 ABLATION STUDY

We include quality and quantity results of model with different setting in Fig. 6 and Table 2. We employ a noise strength of 0.5 for more apparent comparison. The model (a), which uses uni-directional attention (see Fig. 1 (c)) fails to be consistent with the input from the first frame. Columns (b), (c) and (d) are trained with our uni-directional attention with warmup (red square of Fig. 1 (d)). But the model in (b) fills the warmup area with further predecessor frames instead of initial frames (see also Fig. 4). This does improve the output for the first frame (as the predecessor frames happen to be initial), but leads to deviation from the spatial structure of the input in later frames. In column (c) we do use the warmup area properly, but omit the depth prior. The identity of the subject is now maintained better, but several details in the background, such as the highlighted table region are still inconsistent with the input. Only our full method (column (d)) maintains consistency beyond initial frames. Table 2 confirms these findings: Without the depth prior, configurations A, B and C fail to be structurally consistent with the input. And with further predecessor frames instead of initial frames in the warmup area at inference time, configuration B does not achieve as much temporal

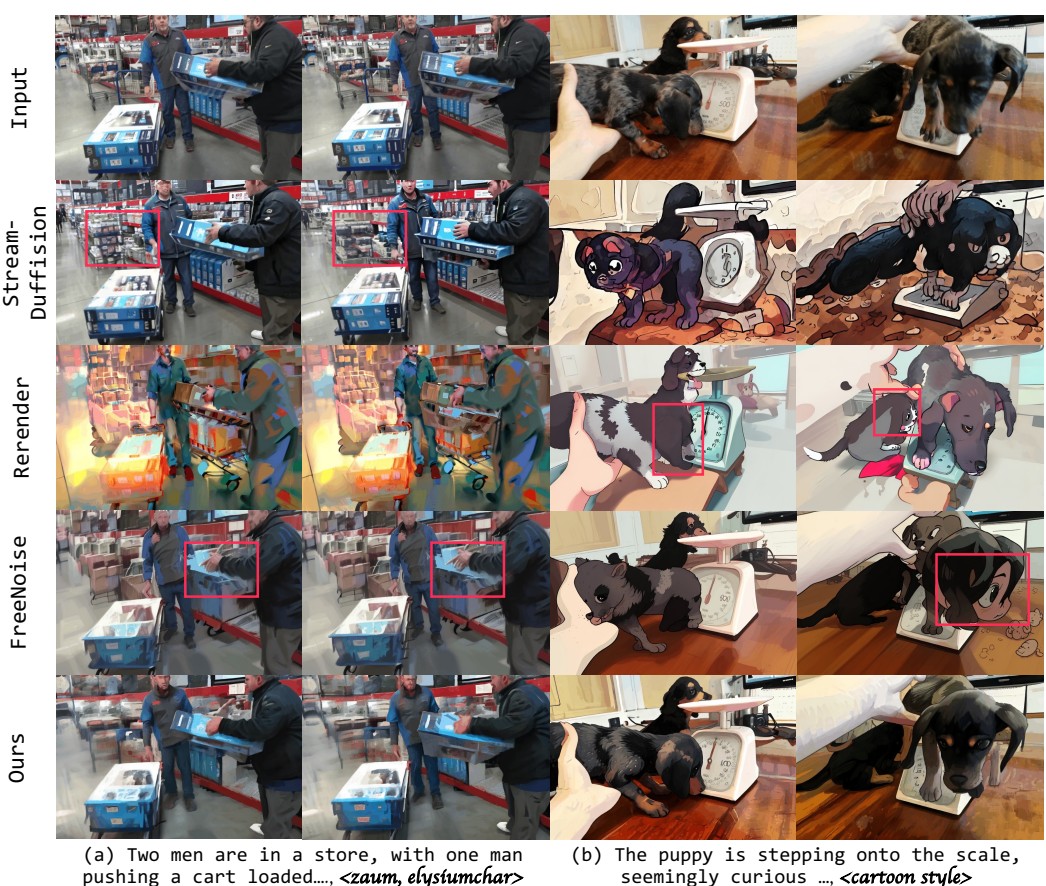

(a) Two men are in a store, with one man pushing a cart loaded…., *<zaum, elystumchar>*

(b) The puppy is stepping onto the scale, seemingly curious …, *<cartoon style>*

Figure 5: We compare the output quality of our method to a number of previous approaches: (a) shows temporally adjacent frames, while (b) shows frames temporally further apart. While our method preserves the spatial structure of the input well, producing the desired output styles, previous methods tend to change even the semantic content of the frames. See more discussions in Section 4.2

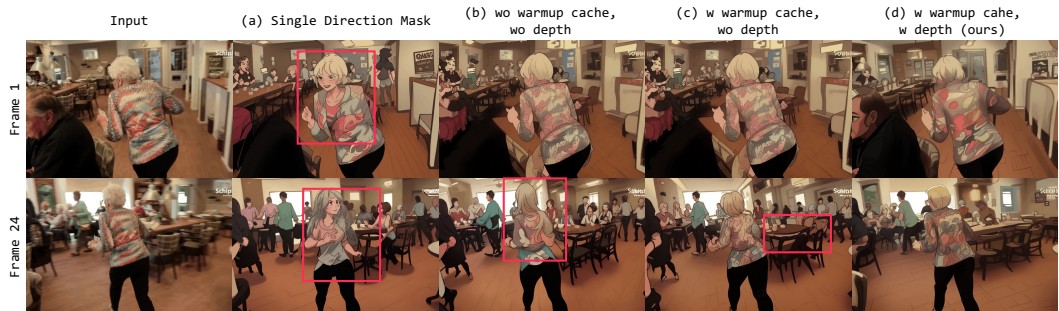

The woman in the colorful shirt is walking towards the table, *<cartoon style>*

Figure 6: In this ablation study, model (a) was trained with an attention mask like in Fig. 1 (c). Models (b) and (c) were trained with the attention masks like in Fig. 1 (d), but in (b) we filled the warmup slots (see Fig. 4) with further close-by predecessor frames instead of initial frames of the stream. Only (d), our full method, uses the depth prior. The models mostly agree on Frame 1, but all ablated versions deviate from the spatial structure of the input for later frames. More analysis in Section 4.3.

| Method | Structure Consistency | | Temporal Smoothness | | | Latency ↓ |
|---|---|---|---|---|---|---|
| | Depth MSE ↓ | Ours Win Rate ↑ | CLIP Score ↑ | Warp Error ↓ | Ours Win Rate ↑ | |
| StreamDiffusion | 1.72 | 86.55% | 94.35 | 0.1994 | 91.81% | **0.03s** (±0.01) |
| Rerender | 5.50 | 73.89% | 95.05 | 0.1327 | 56.67% | 7.72s (±0.18) |
| FreeNoise | 1.94 | 97.70% | **96.49** | **0.0809** | 94.83% | 58.67s (±0.08) |
| Ours | **1.12** | - | 95.77 | 0.0967 | - | 0.06s (±0.02) |

Table 1: To compare our method to previous work, we averaged scores over 90 sequences from the DAVIS-2017 Pont-Tuset et al. (2017) dataset. Our method scores **highest** in Depth MSE and second in terms of temporal smoothness. However, because FREENOISE is actually unable to produce output frames before having seen a number of future input frames, we had to give it an unfair advantage by having it consume *all* input frames before producing its first output frame, leading to extreme latency and explaining why it can achieve better temporal smoothness than all other methods. More details of the metrics in Section 4.1. Our user study win rates confirm that our method produces the best quality for both aspects (i.e. all win rates over 50%). Only STREAMDIFFUSION, which puts more emphasis on speed than on output quality (see also Fig. 5) can beat our method in terms of latency.

| | Setting | | | Structure Consistency | Temporal Smoothness | |
|---|---|---|---|---|---|---|
| | Train with warmup | Inference with warmup | Use depth | Depth MSE ↓ | CLIP Score ↑ | Warp error ↓ |
| A | × | × | × | 2.29 | 95.43 | 0.0968 |
| B | ✓ | × | × | 2.39 | 95.28 | 0.1125 |
| C | ✓ | ✓ | × | 2.22 | **95.80** | 0.0966 |
| D | ✓ | ✓ | ✓ | **1.67** | 95.78 | **0.0768** |

Table 2: Ablation study of the model design. Our full method (D) achieves the **optimal** in both structure consistency and temporal smoothness warp error. As is to be expected, training with warmup, but filling the warmup area with immediate predecessor frames at test time (B) makes quality worse, but using the warmup area correctly (C) does lead to slight improvements over no warmup at all. The depth prior leads to a strong improvement again (D).

consistency as the others. We also found that removing the warmup cache from configuration D will decrease the temporal smoothness CLIP score by 0.09. The depth prior seems to improve both structure consistency and temporal smoothness a lot, although the temporal CLIP score fails to show that in Table 2. We interpret this failure as a consequence of the content of subsequent frames being usually very similar, such that the CLIP embeddings can be similar despite certain abrupt changes, for example in the background, being present.

As reported in Table 3, omitting our $KV$-cache leads to our method having to re-compute the $K$ and $V$ maps of previous frames multiple times, which dramatically increases the per-frame latency to a degree that is not acceptable in streaming use cases.

## 5    CONCLUSION

We have presented LIVE2DIFF, a method to translate video streams to a desired target style at interactive framerates. Based on our novel unidirectional attention approach, that allows us to reduce computational cost by means of our $KV$-cache, we are able to not only meet the criterion of sufficient framerate, but also outperform previous approaches in terms of consistency with the input video and temporal smoothness. We have thus demonstrated that the

| Method | Latency ↓ |
|---|---|
| wo $KV$-cache | 20.43 (±0.068) |
| Ours | 0.06s (±0.002) |

Table 3: Removing the $KV$-cache from our method drastically increases latency.

unidirectional temporal attention mode, that is an important component of state of the art LLMs, can beneficially be used for the editing of videos as well. A method like ours could be of great use in a number of video streaming use cases, such as in the recent trend of "Virtual YouTubers", in which online content producers control stylized virtual avatars and interact with their audience in a live stream.

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

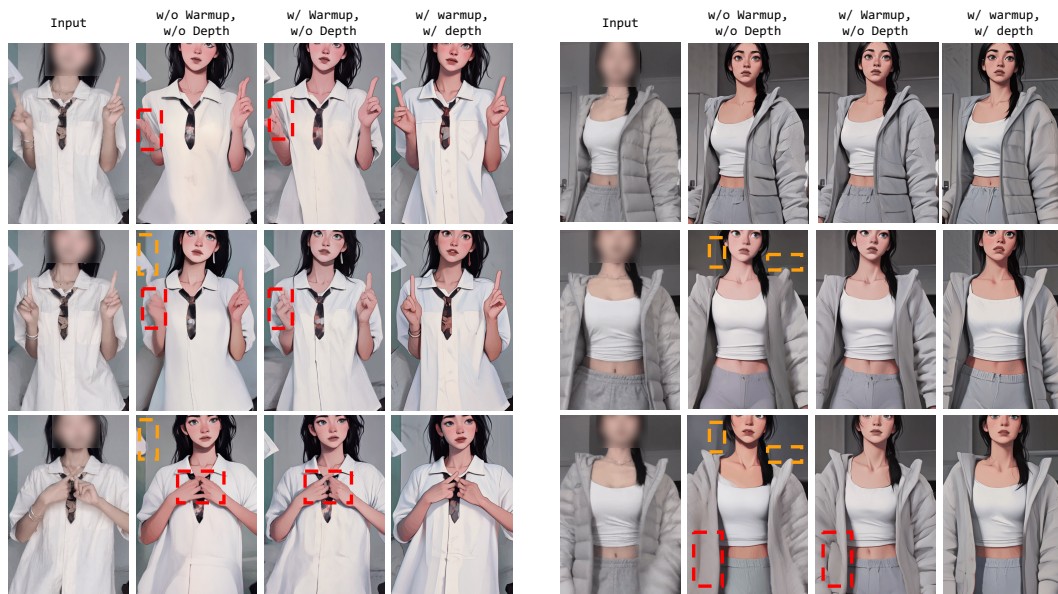

Figure 7: We show a detailed ablation study of our warmup mechanism on two different sequences: Orange boxes denote region with background blurriness without our warmup design, and red boxes denote bad structure consistency region fixed by depth condition. Warmup and depth both play crucial role in our model.

## A APPENDIX

### A.1 ABLATION ON SELECTION OF NUMBER OF WARMUP FRAMES

| Warmup Frames | CLIP Score (↑) | Depth MSE (↓) |
|---|---|---|
| 4 | 95.63 | 2.25 |
| 8 | **95.80** | **2.22** |

Table 4: **Ablation study of selection of warmup frames numbers.**

We ablate the selection warmup frames in Table 4. We experimented with training the model using fewer warm-up frames (without the depth condition), and the results are shown in the table above. Therefore, we ultimately chose 8 (half of the full attention window) as the length for the warm-up frames.

### A.2 MORE ABLATION ON EFFECT OF WARMUP MECHANISM

In Figure Fig. 7, we further analyze the effectiveness of the warmup b bmechanism through additional visualization results. The yellow and red boxes mark background and foreground areas that become blurry if warmup is not used. By utilizing the warmup mechanism, the background blur issue can be mitigated, demonstrating the effectiveness of our design.

### A.3 COMPARISON WITH MORE BASELINE METHODS

We also add more comparisons about FateZero(Qi et al., 2023) and TokenFlow(Geyer et al., 2023) on the DAVIS dataset, as shown in the above table. Compared with our method, those methods achieve better temporal smoothness and worse structure consistency. Both those methods contain bidirectional interaction between all input frames (e.g., spatial-temporal self-attention with middle frame in FateZero(Qi et al., 2023) and tokenflow propagation in TokenFlow(Geyer et al., 2023)), and the inference latency of those methods are unacceptable for streaming data.

| Methods | CLIP Score ($\uparrow$) | Depth MSE ($\downarrow$) | Latency ($\downarrow$) |
|---|---|---|---|
| FateZero | 96.09 | 2.04 | 8.73s |
| TokenFlow | **97.55** | 2.39 | 5.93s |
| Ours | 95.77 | **1.12** | **0.07s** |

Table 5: **Comparison with more baseline methods.**

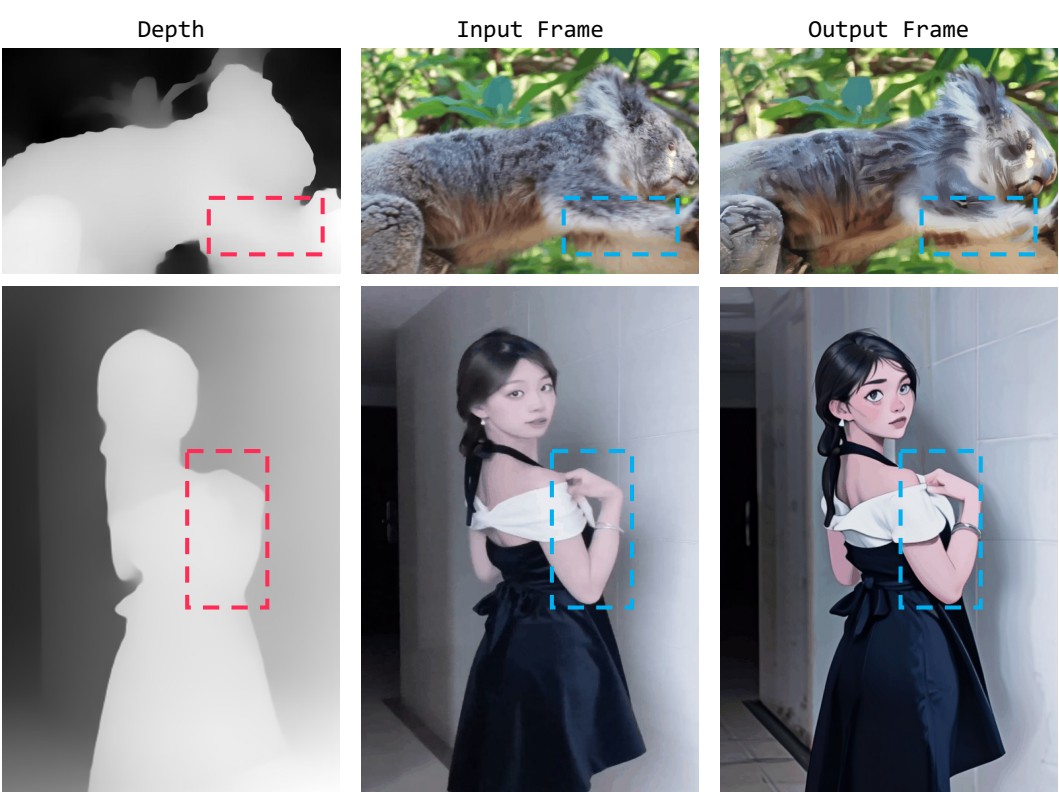

Figure 8: We present two example sequences for which the depth estimation gives bad results (red boxes). We find our method to be quite robust to such imperfections (see blue boxes).

### A.4 ROBUSTNESS TO STRUCTURE PRIOR

We add more visualization results about impact of structure prior. Fig. 8 shows few failure cases of depth estimation (red boxes). Nevertheless our method still maintains especially the structure of the hand (see second sequence). This demonstrates that our method is robust to imperfect depth estimation and is not drawing all of its information just from the depth estimate.

### A.5 IMPLEMENTATION DETAILS OF OUR INFERENCE PIPELINE

In this section, we provide the implementation of our temporal self-attention module with $KV$-cache. We also describe how we apply streaming inference.

$KV$-**cache.** In model initialization, we pre-compute the shape of the $KV$-cache for each temporal self-attention module. For temporal attention with max window size $L$, and input feature size $H \times W \times C$, for $T$ denoising steps, the shape of the $KV$-cache should be $(T, H \times W, L, C)$, see Listing 1.

Listing 1: $KV$-cache sizing

```
def set_cache(T, H, W, L, C):
    k_cache = zeros(T, H * W, L, C)
    v_cache = zeros(T, H * W, L, C)
```

```
        register_buffer("k_cache", k_cache)
        register_buffer("v_cache", v_cache)
```

Previous video diffusion modelsGuo et al. (2023); Chen et al. (2024); Wang et al. (2023b) apply absolute positional encoding `PE`, which is added to the input features before the mapping layers `to_q`, `to_k`, `to_v`, which can be formulated as

$$Q = \texttt{to\_q}(\texttt{PE} + \texttt{feat})$$
$$K = \texttt{to\_k}(\texttt{PE} + \texttt{feat})$$
$$V = \texttt{to\_v}(\texttt{PE} + \texttt{feat})$$

We thus cannot directly cache `K`, `V` since they contain positional information. Instead, we pre-compute `to_q(PE)`, `to_k(PE)`, `to_v(PE)` (see Listing 2), and cache only `to_k(feat)`, `to_v(feat)`.

Listing 2: We precompute `to_k(PE)`, `to_v(PE)`.

```
def prepare_pe_buffer():
    pe_full = pos_encoder.pe
    q_pe = F.linear(pe_full, to_q.weight)
    k_pe = F.linear(pe_full, to_k.weight)
    v_pe = F.linear(pe_full, to_v.weight)

    register_buffer("q_pe", q_pe)
    register_buffer("k_pe", k_pe)
    register_buffer("v_pe", v_pe)
```

In the warmup stage, we use bi-directional attention over all warmup frames, and cache their `K`/`V`, see Listing 3.

Listing 3: Warmup frames are processed with *bi*-directional attention.

```
def temporal_self_attn_warmup(feat, timestep):
    """
    feat: [HW, L, C_in]
    """
    q = to_q(feat)   #  [HW, L, C]
    k = to_k(feat)   #  [HW, L, C]
    v = to_v(feat)   #  [HW, L, C]

    # cache warmup frames before positional encoding
    k_cache[timestep, :, :warmup_size] = k
    v_cache[timestep, :, :warmup_size] = v

    pe_idx = list(range(k.shape[1]))

    pe_q = q_pe[:, pe_idx]
    pe_k = k_pe[:, pe_idx]
    pe_v = v_pe[:, pe_idx]

    q_full = q + pe_q
    k_full = k + pe_k
    v_full = v + pe_v

    # do not use attention mask
    feat = scaled_dot_product_attention(
        q_full,
        key_full,
        value_full,
        attention_mask=None)

    feat = to_out(feat)
    return feat
```

During streaming inference we process up to $T$ samples with different noise levels at once. For each frame we write to and read from the $KV$-cache corresponding to its noise level, and add the mapped

positional information. At the beginning of the stream, the length of context window is incrementally approaching the max window size $L$. We pass an attention mask to specify which token should take part in attention. For details see Listings 4 and 5.

Listing 4: Our implementation of streaming inference uses the uni-directional attention approach, see Fig. 1 (d).

```python
def temporal_self_attn_streaming(feat, attn_mask):
    """
    feat: [THW, L, C_in]
    attn_mask: [T, L], 0 for attention, -inf for no attention
    """
    q_layer = rearrange(q_layer, "(nhw) f c -> n hw f c", n=T)
    k_layer = rearrange(k_layer, "(nhw) f c -> n hw f c", n=T)
    v_layer = rearrange(v_layer, "(nhw) f c -> n hw f c", n=T)

    # handle prev frames, roll back
    k_cache[:, :, warmup_size:] = k_cache[:, :, warmup_size:] \
                                    .roll(shifts=-1, dims=2)
    v_cache[:, :, warmup_size:] = v_cache[:, :, warmup_size:] \
                                    .roll(shifts=-1, dims=2)
    # write curr frame
    k_cache[:, :, -1:] = k_layer
    v_cache[:, :, -1:] = v_layer

    k_full = k_cache
    v_full = v_cache

    # attn_mask:
    #    [[0, 0, 0, 0, -inf, -inf, 0, 0],
    #     [0, 0, 0, 0, -inf, -inf, -inf, 0]]
    # then pe for each element shoule be
    #    [[0, 1, 2, 3, 3, 3, 4, 5],
    #     [0, 1, 2, 3, 3, 3, 3, 4]]
    kv_idx = (attn_mask == 0).cumsum(dim=1) - 1  # [T, L]
    q_idx = kv_idx[:, -q_layer.shape[2]:]  # [T, 1]

    # [n, window_size, c]
    pe_k = concatenate([
        k_pe.index_select(1, kv_idx[idx])
        for idx in range(T)], dim=0)
    pe_v = concatenate([
        v_pe.index_select(1, kv_idx[idx])
        for idx in range(T)], dim=0)
    pe_q = concatenate([
        q_pe.index_select(1, q_idx[idx])
        for idx in range(T)], dim=0)

    q_layer = q_layer + pe_q.unsqueeze(1)
    k_full = k_full + pe_k.unsqueeze(1)
    v_full = v_full + pe_v.unsqueeze(1)

    q_layer = rearrange(q_layer, "n hw f c -> (n hw) f c")
    k_full = rearrange(k_full, "n hw f c -> (n hw) f c")
    v_full = rearrange(v_full, "n hw f c -> (n hw) f c")

    attn_mask_ = attn_mask[:, None, None, :].repeat(
        1, h * w, q_layer.shape[1], 1)
    attn_mask_ = rearrange(attn_mask_, "n hw Q KV -> (n hw) Q KV")
    attn_mask_ = attn_mask_.repeat_interleave(heads, dim=0)

    feat = scaled_dot_product_attention(
        q_full,
        key_full,
        value_full,
```

```
        attention_mask=attention_mask_)

    feat = to_out(feat)
    return feat
```

Listing 5: During inference we process $T$ frames simultaneously to make full use of GPU parallelization.

```
def streaming_v2v(frame):
    """
    frame: [1, 3, H, W]
    """
    latent = vae.encode(frame)
    depth_latent = vae.encode(depth_detector(frame))
    noisy_latent = add_noise(latent)  # add noise based on SDEdit
    if prev_latent is None:
        prev_latent = randn([T-1, ch, h, w])
    if attn_mask is None:
        attn_mask = zeros(T, L)
        attn_mask[:, :warmup_size] = 1
        attn_mask[0, -1] = 1  # curr frame participate attention
        attn_mask.masked_fill_(attn_mask == 0, float("-inf"))

    latent_batch = concatenate([noisy_latent, prev_latent])
    noise_pred = UNet(latent_batch, depth_latent,
                      t, text_embedding, attn_mask)
    denoised_latent = scheduler.step(noise_pred, latent_batch, t)

    out_latent = denoised_latent[-1]
    prev_latent = denoised_latent[1:]

    out_frame = vae.decode(out_latent)
    return out_frame
```

| Models | Trigger Words |
|---|---|
| Flat-2D Animerge [1] | *cartoon style* |
| zaum [2] | *zaum, elysiumchar* |
| vangogh [3] | *Starry Night by Van Gogh, lvngvncnt* |

Table 6: Community models used for evaluation. Each model captures a different target style.

### A.6 EVALUATION

**Models for evaluation.** We use three Dreambooth and LoRA settings for evaluation, the model name and trigger words are shown in Table 6. For the evaluation, we use the trigger word as the prefix of our prompt.

**User study.** Our user study involved 31 participants. The video clips were the same as those used in the quantitative evaluation. Fig. 9 illustrates the user interface of our user study system: Participants are shown the input video as reference, as well as an output from our method and an output from one random baseline method. They are asked to select the output with better temporal smoothness and structure consistency to the input. For each baseline method, we compute the win rate of our method as

$$\text{ours\_win\_rate} = 1 - \frac{\text{baseline\_voted}}{\text{baseline\_shown}} \tag{3}$$

---

[1]https://civitai.com/models/35960/flat2danimerge

[2]https://civitai.com/models/16048/or-disco-elysium-style-lora

[3]https://civitai.com/models/91/van-gogh-diffusion

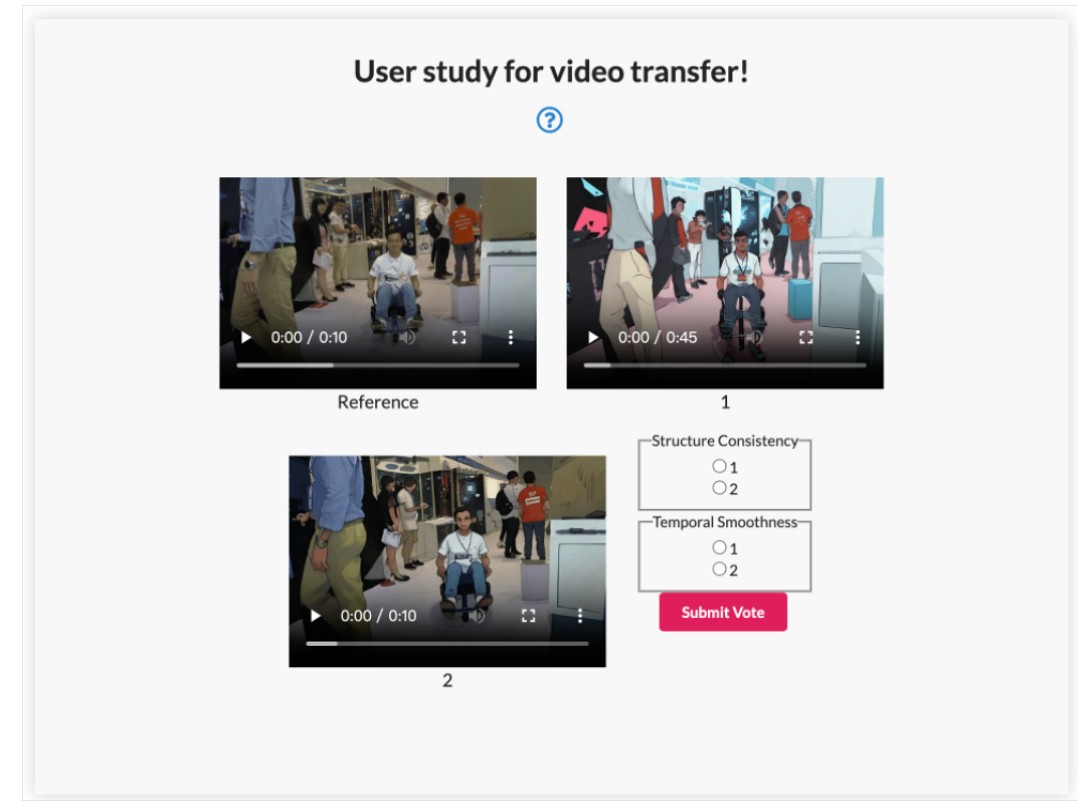

Figure 9: In our user study, the participant is given triplets of videos: The "Reference" is the input video, that videos 1 and 2 should be structurally consistent with, in addition to being temporally smooth. For each of these two aspects the user chooses which of the two videos fulfills this aspect best.

**Data captioning.** We caption the DAVIS dataset with CogVLMWang et al. (2023c), which is a state-of-the-art visual language model. For each video clip, we feed the middle frame together with the following prompt:

> *Please caption the given image. The caption should focus on the main object in image and describe the motion of the object.*

## A.7 APPLICATION

Fig. 10 shows another application of our method, demonstrating its potential in virtual-liver cases. We transfer the input videos to different styles at 10 - 15 FPS on an NVIDIA RTX 4090.

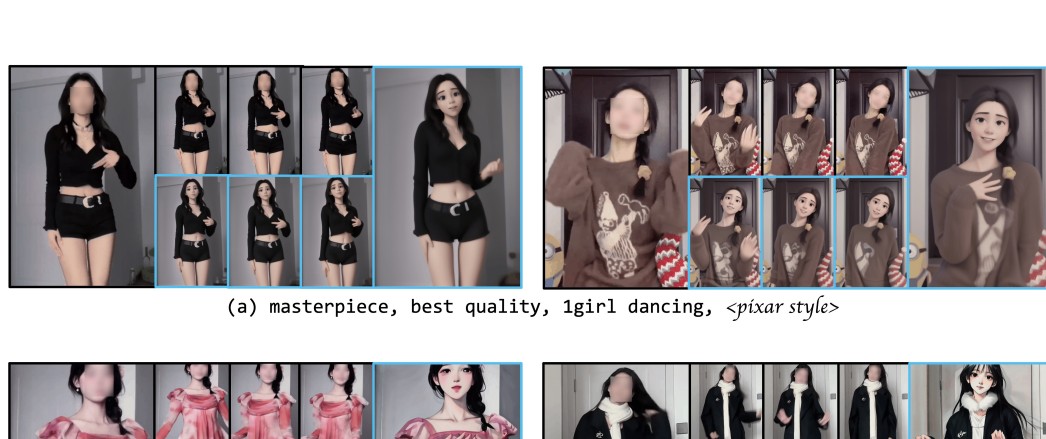

(a) masterpiece, best quality, 1girl dancing, *<pixar style>*

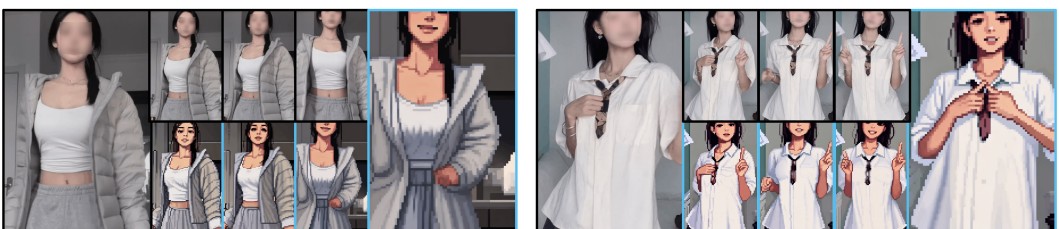

(b) 1girl, dancing,
*<shukezouma, negative space, shuimobysim, official art, extremely detailed CG,unity 8k wallpaper,chinese ink painting>*,

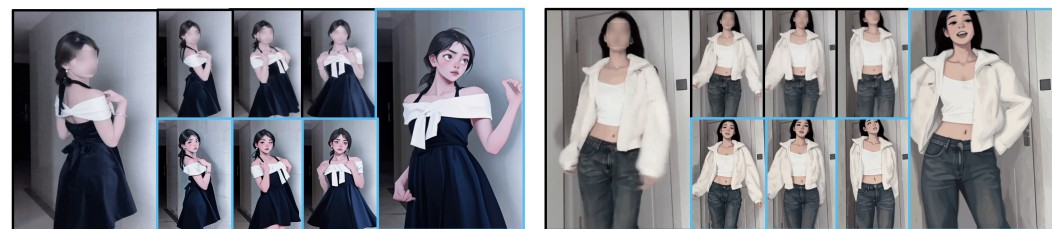

(c) masterpiece, best quality, 1girl dancing, *<pixart>*,

(c) masterpiece, best quality, 1girl dancing, *<cartoon>*,

Figure 10: Our method translates an input video stream (black boundaries) into an output video stream (blue boundaries) that conforms to a desired target style. Each prompt is composed of the caption obtained for the input video (see Appendix A.6 followed by target style.

