# OpenReview forum: "Live2Diff: Live Stream Translation via Uni-directional Attention in Video Diffusion Models"
_ICLR.cc/2025/Conference — ICLR 2025 Conference Withdrawn Submission_

### Official Review · Reviewer_JBxd · 2024-10-22

**Soundness:** 2
**Presentation:** 3
**Contribution:** 2
**Rating:** 5
**Confidence:** 3

**Summary:**

The paper proposes Live2Diff, a video diffusion model that utilizes uni-directional temporal attention as well as additional warmup and depth information for streaming video translation. In specific, the uni-directional attention is conditioned on a sliding window of a few preceding frames, and the warmup contains attention from a small number of frames at the beginning of the video. To achieve real-time inference, Live2Diff uses three techniques: First, it replaces vanilla diffusion models with LCM-LoRA to reduce the number of denoising steps. Second, it uses a KV-cache during the implementation to avoid redundant computation. Third, it designs a pipelined denoising approach to accelerate model inference. In the experiments, the author(s) compare Live2Diff with three video translation methods, including StreamDiffusion, Rerender, and FreeNoise, and demonstrate that Live2Diff achieves higher CLIP scores and lower warp error while requiring lower latency.

**Strengths:**

1.	The paper is well-written and easy-to-understand.
2.	The proposed method achieves high efficacy (16 FPS on single RTX4090) for streaming video translation.
3.	The ablation experiments are well-studied and demonstrate the necessity of each component in Live2Diff.

**Weaknesses:**

1.	I have some doubts about the novelty in this paper. In specific, just like the authors mentioned, several works have adopted an autoregressive way for long video generation, e.g. ART-V, SEINE, Gen-L-Video, Reuse and Diffuse. The idea of incorporating a small number of frames in the beginning is also not new [1]. The author(s) claim their major contributions of proposing uni-directional attention with warmup. However, it seems naive to me that the author(s) simply replace bidirectional attentions with uni-directional attentions because of the problem setting, where the target frame shall not have seen the later frames and can only use the information from preceding frames. Regarding warmup, it can be seen in Table 2 that warmup does not provide significant improvements. (Row A, B, and C have similar values.) Instead, the improvement comes from using additional depth priors instead of model architecture.
2.	The selected baselines are somehow outdated. Similar work, such as StreamV2V (Open source, Arxiv, May 2024), that also targets real-time streaming video should be included for comparison. State-of-the-art video translation methods, like CoDeF and FlowVid, which may have better temporal consistency, editing performance, but lower throughput, should also be compared. In specific, StreamV2V lies in exactly the same scope as the paper, and they compare with both CoDeF and FlowVid, both of which outperform the selected baselines, i.e. Rerender and FreeNoise, with much better temporal consistency.
3.	The literature review is not complete enough. There are several missing methods for video processing. To my best knowledge, existing methods can mainly be classified into four categories: propagation-based, representation-based, canonical-based, and attention-based. These methods use different strategies to improve temporal consistency, like propagation across frames [2,3], video deformation [4,5], transferring video to canonical space [6,7], or using attention mechanisms. It is acceptable that the paper only focuses on attention-based methods, which is much closer to the scope, but there are still many missing approaches that use diffusion models for video generation/editing. Some of them are zero-shot methods that do not require model training, e.g. Controlvideo, Tokenflow, Control-A-Video,  VideoControlnet, Text2video-zero. Some of them further enhance temporal consistency with cross-attention, e.g. Tune-A-Video, Text2Video-Zero, FateZero, Vid2Vid-Zero. Some works propose training for video editing, e.g. Imagen Video, Make-A-Video. In short, I suggest the author(s) provide more comprehensive literature review in Section 2.
4.	The temporal consistency is bad in some examples demonstrated. (Comparison 1, 2, 3, 6) To my knowledge, this is because the backgrounds in these examples contain more 'high-frequency' details. In other words, they are more complicated, compared to other 'dancing human' videos in indoor scenes with simple background. Since the paper has incorporated the depth priors, is it possible to first differentiate the static (background) and dynamic objects (human subjects, for example) in the video, and then do a simple masking techniques to avoid temporal inconsistency in the static background?


[1] W Harvey et al., Flexible Diffusion Modeling of Long Videos, NeurIPS 2022.

[2] O Jamriška, Stylizing video by example. ACM TOG 2019.

[3] X Wang et al., Learning Correspondence From the Cycle-Consistency of Time, CVPR 2019.

[4] V Ye et al., Deformable Sprites for Unsupervised Video Decomposition, CVPR 2022.

[5] Y Kasten et al., Layered neural atlases for consistent video editing, ACM TOG 2021.

[6] H Ouyang et al., CoDeF: Content Deformation Fields for Temporally Consistent Video Processing, CVPR 2024

[7] T Chen et al., NaRCan: Natural Refined Canonical Image with Integration of Diffusion Prior for Video Editing, NeurIPS 2024.

**Questions:**

1.	Figure 3 and Figure 6 do not demonstrate strong initiatives in using depth priors. Since the baselines may not have depth priors, I am curious about the qualitative comparison between baselines, uni-directional attention with warmup, and uni-directional attention with warmup and depth.
2.	Could you provide the video results of Figure 3? It seems that the full model with Figure 3(f), i.e. full model with depth, has worse temporal consistency than Figure 3(e), i.e. full model without depth, specifically at the regions of human clothes and face. I expect serious jittering artifacts in Figure 3(f).
3.	Could you provide the comparison with more baselines, especially StreamV2V?
4.	It seems to me that all the generated results are very similar to the input videos, and the prompt must align with the input videos. Is it possible for Live2Diff to do other editing tasks besides style transfer, for example, changing the color of human clothes?
5. See weakness.

**Details Of Ethics Concerns:**

The video-to-video translation may be used to generate fake videos for particular human subjects.

---

### Official Review · Reviewer_z2aW · 2024-10-31

**Soundness:** 3
**Presentation:** 1
**Contribution:** 2
**Rating:** 3
**Confidence:** 4

**Summary:**

this paper presents a comprehensive method for streaming video processing, leveraging on a unidirectional attention (with warmup frames and context window) of video frames and pipelined denoising (proposed in streamdiffusion) with kv caching.  for streaming video processing application, depth information is extracted and proved to be very important for good performance.

**Strengths:**

the problem this paper is working on is important, as streaming and low-latent video processing and generation has broad applications. the approach proposed by this paper makes sense as pipelined processing and temporal relation modeling are both important for decent performance under tight latency constraints.

**Weaknesses:**

overall, the writing of this paper is not good (e.g. "Absolute position encoding (i.e., sinusoidal position encoding) to fin before computing Eq. (2), to give the layer access to the temporal position of each feature vector."), and what's more important, the proposed approach lacks technical novelty.
1. the pipelined denoising is firstly proposed in streamdiffusion.
2. uni-directional attention and kv caching are both common in LLM community. the warmup design also lacks insightful analyses.
3. depth information extraction seems to be highly heuristic and specialized to the video to video application.

**Questions:**

1. how does your method work in other application scenarios? for example text to video generation
2. "We measure inference speed as the total amount of time it takes each method to process an input stream of 100 frames" --- but this is not the end to end latency at user side?
3. why is depth information so important?  what will happen if the depth is inaccurate?
4. in Figure 5, the results are in different styles, which may lead to unfair comparisons?

---

### Official Review · Reviewer_6dyc · 2024-11-04

**Soundness:** 3
**Presentation:** 4
**Contribution:** 3
**Rating:** 6
**Confidence:** 3

**Summary:**

This paper introduces a novel approach to real-time video style transfer for streaming applications. The key innovation is adapting unidirectional attention mechanisms (similar to those used in Large Language Models) for video diffusion models, allowing frame-by-frame processing without requiring future frames. Additionally, they implemented warmup and KV-cache strategies.

**Strengths:**

- The paper is well-motivated, addressing an important practical problem of live video stream translation.
- The proposed unidirectional attention mechanism is a natural solution inspired by successful LLM architectures.
- Clear explanation of the architecture and mechanisms.
- Multiple evaluation metrics and user study results including structural consistency and temporal smoothness.

**Weaknesses:**

**Limited Analysis of Dynamic Scenes**

- The paper lacks a comprehensive evaluation of videos containing rapid motion or frequent scene changes. The authors are recommended adding comparative experiments between the warmup approach and simple unidirectional attention for several scenarios: videos with fast camera movements, videos with quick scene transitions, and action sequences with rapid object motion. In addition, I'm concerned if the warmup window might become useless or perform worse than simple unidirectional attention when the content is no longer relevant to the first few frames.

**Depth Predictor Integration & Evaluation**

- The current depth Mean Square Error (MSE) metric creates an unfair comparison between methods. To address this limitation, I suggest adding structural similarity metrics between frames that are independent of depth, such as SSIM and LPIPS. Furthermore, evaluations should be conducted both with and without depth conditioning to ensure a more equitable comparison.

**Visual Quality Issues**
- Several visual artifacts are noticeable in the results presented in the supplementary material. I suggest including detailed analyses of these failure cases to better understand the method's limitations and potential areas for improvement.

**Questions:**

- How does the model perform on videos with rapid scene changes or fast motion where early warmup frames might become less relevant?
- Are there justifications for using the depth MSE metric to measure structural consistency? The use of an off-the-shelf depth predictor can be an advantage.

---

### Note · Authors · 2024-11-23

I have read and agree with the venue's withdrawal policy on behalf of myself and my co-authors.